# Bipartite Stochastic Block Models with Tiny Clusters

Stefan Neumann

University of Vienna
Faculty of Computer Science
Vienna, Austria
`stefan.neumann@univie.ac.at`

## Abstract

We study the problem of finding clusters in random bipartite graphs. We present a simple two-step algorithm which provably finds even tiny clusters of size $O(n^\varepsilon)$, where $n$ is the number of vertices in the graph and $\varepsilon > 0$. Previous algorithms were only able to identify clusters of size $\Omega(\sqrt{n})$. We evaluate the algorithm on synthetic and on real-world data; the experiments show that the algorithm can find extremely small clusters even in presence of high destructive noise.

## 1  Introduction

Finding clusters in bipartite graphs is a fundamental problem and has many applications. In practice, the two parts of the bipartite graph usually correspond to objects from different domains and an edge corresponds to an interaction between the objects. For example, paleontologists use biclustering to find co-occurrences of localities (left side of the graph) and mammals (right side of the graph) [13]; bioinformaticians want to relate biological samples and gene expression levels [10]; in an online shop setting, one wants to find clusters of customers and products.

Discovering clusters in bipartite graphs has been researched in many different settings. However, most of these algorithms were heuristics and do not provide theoretical guarantees for the quality of their results. This was recently addressed by Xu et al. [27] and Lim et al. [17] who initiated the study of biclustering algorithms with formal guarantees. They considered random bipartite graphs and proved under which conditions their algorithms can recover the ground-truth clusters.

In this paper, we consider a standard random graph model and propose a simple two-step algorithm which provably discovers the ground-truth clusters in bipartite graphs: (1) Cluster the vertices on the left side of the graph based on the similarity of their neighborhoods (Section 3). (2) Infer the right side clusters based on the previously discovered left clusters using degree-thresholding (Section 4).

Our algorithm allows to recover even tiny clusters of size $O(n^\varepsilon)$, where $n$ is the number of vertices on the right side of the graph and $\varepsilon > 0$. Previously, existing algorithms could only discover clusters of size $\Omega(\sqrt{n})$. Note that finding tiny clusters is of high practical importance. For example, in an online shop with millions of products ($n \geq 10^6$), finding only clusters of at least a thousand products ($\sqrt{n} \geq 10^3$) is not very interesting. One would want the product clusters to be much smaller.

The formal guarantees of our algorithm are provided at the end of this section. From a high-level point of view, the algorithm can be seen as a way to leverage formal guarantees for mixture models and clustering algorithms into biclustering algorithms with formal guarantees. This partially explains why heuristics such as "apply $k$-means to both sides of the graph" are very successful in practice.

Finally, we implement a version of the proposed algorithm (Section 5) and we evaluate it on synthetic and on real-world data. The experiments show that in practice the algorithm can find extremely small clusters and it outperforms all algorithms we compare with (Section 6).

**Bipartite Stochastic Block Models.** We now introduce *bipartite stochastic block models (SBMs)* which we will be using throughout the paper. Let $G = (U \cup V, E)$ be a bipartite graph with $m$ vertices in $U$ and $n$ vertices in $V$; we call $U$ the *left side* of $G$ and $V$ the *right side* of $G$.

The left side $U$ is partitioned into clusters $U_1, \ldots, U_k$, i.e., $U_i \cap U_j = \emptyset$ for $i \neq j$ and $\bigcup_i U_i = U$. For $V$ there are clusters $V_1, \ldots, V_k$ with $V_i \subseteq V$; we do not assume that the $V_j$ are disjoint or that their union equals $V$. The $U_i$ are the *left clusters* of $G$ and the $V_j$ are the *right clusters* of $G$.

Fix two probabilities $p > q \geq 0$. For any two vertices $u \in U_i$ and $v \in V_i$, insert an edge with probability $p$; for $u \in U_i$ and $v \notin V_i$, insert an edge with probability $q$.

The algorithmic task for bipartite SBMs is as follows. Given parameters $k$, $p$, $q$ and a graph $G$ generated in the previously described way, recover all clusters $U_i$ and $V_j$.

**Main Theoretical Results.** We propose the following simple algorithm:

1. Recover the clusters $U_i$ by clustering the vertices in $U$ according to the similarity of their neighborhoods (see Section 3).
2. For each recovered $U_i$, set $V_i$ to all vertices with "many" neighbors in $U_i$ (see Section 4).

To state the formal guarantees of the proposed algorithm we require two parameters. We let $\ell$ be the size of the smallest cluster on the left side, i.e., $\ell = \min_i |U_i|$. Furthermore, let $\delta$ denote the smallest difference between any two clusters on the right side; more formally, $\delta = \min_{i \neq j} |V_i \Delta V_j|$, where $V_i \Delta V_j = (V_i \setminus V_j) \cup (V_j \setminus V_i)$ is the symmetric difference of $V_i$ and $V_j$.

We now state the main result of this paper. In the theorem, $D(p \parallel q)$ denotes the Kullback–Leibler divergence of Bernoulli random variables with parameters $p, q \in [0, 1]$, i.e., $D(p \parallel q) = p \lg(\frac{p}{q}) + (1 - p) \lg(\frac{1-p}{1-q})$.

**Theorem 1.** *Suppose $\sigma^2 = \max\{p(1-p), q(1-q)\} \geq \lg^6 n/n$. There exist constants $C_1, C_2$ such that if $\ell \geq C_1 \lg n/D(p \parallel q)$ and*

$$\frac{(p-q)^2}{\sigma^2} > C_2 k \frac{n + m \lg m}{\ell \delta}, \tag{1}$$

*then there exists an algorithm which on input $G$, $k$, $p$ and $q$ returns all clusters $U_i$ and $V_i$. The algorithm succeeds with high probability.*

To give a better interpretability of the theorem, consider its two main assumptions: (1) The condition $\ell \geq C_1 \lg n/D(p \parallel q)$ is necessary so that the vertices in $V_i$ have sufficiently many neighbors in $U_i$. (2) To get a better understanding of Equation 1, consider the case where $m = \Theta(n)$, $k = O(1)$, and $p, q$ are constants. Also, ignore logarithmic factors. We obtain a smooth tradeoff between $\delta$ and $\ell$: The inequality in Equation 1 is satisfied if $\delta = \Theta(n^\varepsilon)$ and $\ell = \Theta(n^{1-\varepsilon})$. That is, if the right clusters are very small or similar ($\delta$ is small), the algorithm requires larger clusters on the left side ($\ell$ must be large). On the other hand, if the right clusters are very large and dissimilar (large $\delta$), the algorithm requires only very small left clusters (small $\ell$ suffices). More generally, if $p - q = \Theta(n^{-C})$ and $p \gg q$, the algorithm requires $\ell\delta = \Theta(n^{1+C})$.

The fact that the algorithm can recover clusters of size $O(n^\varepsilon)$ is interesting since previous algorithms required $\min\{\ell, \delta\} = \Omega(\sqrt{n})$ (see Section 2). Furthermore, the lower bounds of Hajek, Wu and Xu [15] show that breaking the $\Omega(\sqrt{n})$ barrier is *impossible* in general graphs. Hajek et al. also provide lower bounds in the bipartite setting which show that one cannot find biclusters of size $k \times k$ for $k = o(\sqrt{n})$. We bypass this lower bound through the previously discussed smooth tradeoff between $\ell$ and $\delta$. We conjecture that the tradeoff we obtain is asymptotically optimal.

As a side result we also study the setting in which the algorithm only obtains an *approximate* clustering of the left side of the graph. We show that when the approximation of the left clusters is of good enough quality, then the right clusters can still be recovered *exactly*. We also observe this behavior in our experiments in Section 6. We provide the result in the supplementary material.

**Experimental Evaluation.** We implemented a version of the algorithm from Theorem 1 and present the practical details in Section 5. The experimental results are reported in Section 6. In the experiments, our main focus will be to verify whether, in practice, the algorithm can find the small clusters that the theoretical analysis promised.

On synthetic data, the experiments show that, indeed, the algorithm finds tiny clusters even in the presence of high destructive noise and it outperforms all methods that we compare against.

The algorithm is also qualitatively evaluated on real-world datasets. On these datasets it finds clusters which are interesting and which have natural interpretations.

## 2 Related Work

**Stochastic Block Models (SBMs).** During the last years, many papers on SBMs have been published. We only discuss bipartite SBMs here and refer to the survey by Abbe [1] for other settings.

Lim, Chen and Xu [17] study the biclustering of observations with general labels. When constrained to only two labels, their results provide a bipartite SBM. However, in the bipartite SBM case, [17] has two drawbacks compared to the results presented here: (1) The data-generating process in [17] rules out certain nested structures of the sets $V_i$. E.g., [17] does not allow to have clusters $V_1, V_2, V_3$ such that $V_3 = V_1 \cup V_2$. (2) The main result of [17] relies on a notion of *coherence*, which measures how difficult the structure of the clusters is to infer. Due to this dependency on coherence, the results of this paper and [17] are only partially comparable. In case of a constant number of clusters or "worst-case coherence", though, the algorithm of [17] only works if both $\ell$ and $\delta$ are $\Omega(\sqrt{n})$.

Zhou and Amini [28] study spectral methods for bipartite SBMs. [28] considers a more general connectivity structure and obtains sharper bounds for the recovery rates than in this paper. However, in [28] the clusters $V_i$ cannot overlap and, hence, the results of this paper and [28] are incomparable.

Abbe and Sandon [2, 3] and Gao et al. [14] study optimal recovery for SBMs in general graphs. Their results apply to bipartite graphs with a constant number of overlapping communities of linear size. Zhou and Amini [29] improve these results for bipartite SBMs under a broader range of parameters.

One can use the result of McSherry [18] to recover the clusters of a bipartite graph but this has two caveats: (1) It does not allow the $V_i$ to overlap. (2) Both $\ell$ and $\delta$ must be of size $\Omega(\sqrt{n})$.

Florescu and Perkins [11] provided an SBM for bipartite graphs with two linear-size communities on each side of the graph. Xu et al. [27] consider a biclustering setting with clusters of size $\Omega(n)$.

**Boolean Matrix Factorization (BMF).** Another way to find clusters in bipartite graphs is BMF. BMF takes the biadjacency matrix $D \in \{0, 1\}^{m \times n}$ of a bipartite graph and finds factor matrices $L \in \{0, 1\}^{m \times k}$ and $R \in \{0, 1\}^{k \times n}$ such that $D \approx L \circ R$, where $\circ$ is the Boolean matrix-matrix-product. In other words, BMF tries to approximate $D$ with a Boolean-rank $k$ matrix. The interpretation is that the columns of $L$ contain the left clusters and the rows of $R$ contain the right clusters. This setting is more general than the one presented in this paper as it allows the clusters $U_i$ to overlap.

BMF was studied from applied [20, 21, 24–26] and also from theoretical [5, 7, 12] perspectives. Section 6 provides an experimental comparison of BMF algorithms and the algorithm from this paper.

## 3 Recovering the Left Clusters

We describe how the clusters $U_i$ can be recovered. Our approach is to cluster the vertices $u \in U$ according to the similarity of their neighborhoods in $V$. The intuition is that if two vertices $u$ and $u'$ are in the same cluster $U_i$, they should have relatively many neighbors in common (those in $V_i$). On the other hand, if $u$ and $u'$ are from different clusters $U_i$ and $U_j$, their neighbors should be relatively different (as $V_i \Delta V_j$ is supposed to be large).

Technically, we will apply mixture models. We use the result by Mitra [22] since it is simple to state. We could as well use other mixture models such as the one by Dasgupta et al. [9] or clustering algorithms such as Kumar and Kannan [16], Bilu and Linial [6] or Cohen-Addad and Schwiegelshohn [8]. The different methods come with different assumptions on the data.

### 3.1 Mixture Models and Mitra's Algorithm

**Mixture Models on the Hypercube.** From a high-level point of view, the question of mixture models is as follows: Given samples from different distributions, cluster the samples according to which distributions they were sampled from. We will now present the formal details behind this.

Let there be $k$ probability distributions $D_1, \ldots, D_k$ in $\{0,1\}^n$ and denote the mean of $D_r$ as $\mu_r \in [0,1]^n$. Let $\sigma^2$ be an entry-wise upper bound on all $\mu_r$, i.e., $\mu_r(i) \leq \sigma^2$ for all $r = 1, \ldots, k$ and $i = 1, \ldots, n$. For each distribution $D_r$ define a weight $w_r > 0$ such that $\sum_r w_r = 1$.

From each distribution $D_r$, create $w_r m$ samples and denote the set of these samples as $T_r$. In total we obtain $m$ samples and denote the set containing all samples as $T$, i.e., $T = \bigcup_r T_r$.

The algorithmic problem in mixture models is as follows. Given $T$ and $k$, find a partition $P_1, \ldots, P_k$ of the samples in $T$ such that $\{T_1, \ldots, T_k\} = \{P_1, \ldots, P_k\}$.

**Mitra's Algorithm.** We state the result by Mitra [22] and refer to the supplementary material for more details on the algorithm. We define a matrix $A \in \{0,1\}^{m \times n}$ which has the samples from $T$ in its rows. Thus, by clustering the rows of $A$ we obtain a clustering of $T$. The formal guarantees are stated in the following lemma, where $\|v\|_2 = (\sum_i v_i^2)^{1/2}$.

**Lemma 2** (Mitra [22]). *Suppose $\sigma^2 \geq \lg^6 n/n$. Let $\zeta = \min\{\|\mu_r - \mu_s\|_2^2 : r \neq s\}$ and $w_{\min} = \min_r w_r$. Then there exists a constant $c$ such that if*

$$\zeta > ck\sigma^2 \frac{1}{w_{\min}} \left( \frac{m+n}{m} + \lg m \right),$$

*then on input $A$ and $k$, the output $\{P_1, \ldots, P_k\}$ of Mitra's algorithm satisfies $\{P_1, \ldots, P_k\} = \{T_1, \ldots, T_k\}$ with high probability. That is, the algorithm recovers the clusters $T_r$.*

### 3.2 Proposition and Analysis

Let us come back to our original problem of recovering the left clusters of $G$. To find the left clusters $U_i$, we apply Mitra's algorithm to the rows of the biadjacency matrix $D$ of $G$. Formally, the biadjacency matrix $D \in \{0,1\}^{m \times n}$ of $G$ is the matrix with $D_{uv} = 1$ iff there exists an edge $(u,v) \in G$.

Proposition 3 states under which conditions this approach succeeds.

**Proposition 3.** *Let all variables be as in Section 1. Let $\delta = \min_{i \neq j} |V_i \Delta V_j|$ and $\ell = \min_i |U_i|$. Suppose $\sigma^2 = \max\{p(1-p), q(1-q)\} \geq \lg^6 n/n$. There exists a constant $C$ such that if*

$$\frac{(p-q)^2}{\sigma^2} > Ck \frac{n + m\lg m}{\ell\delta}, \tag{2}$$

*then applying Mitra's algorithm on $D$ returns a partition $\{\tilde{U}_1, \ldots, \tilde{U}_r\}$ of $D$'s rows such that $\{\tilde{U}_1, \ldots, \tilde{U}_r\} = \{U_1, \ldots, U_r\}$ with high probability. That is, the algorithm recovers the left clusters $U_i$ of $G$.*

*Proof.* Observe that $D$ is a matrix arising from a mixture model as discussed earlier: Consider a vertex $u \in U_i$ and its corresponding row $D_u$ in $D$. Then the probability that entry $D_{uv} = 1$ is $p$ if $v \in V_i$ and $q$ if $v \notin V_i$. Furthermore, for two vertices $u, u' \in U_i$ these distributions are exactly the same.

Hence, we view the rows of $D$ as samples from $k$ distributions $D_i$ with distribution $D_i$ corresponding to cluster $U_i$. For each cluster $U_i$, we have $|U_i|$ samples from $D_i$. For the mean $\mu_i$ of $D_i$, we have component-wise $\mu_i(v) = p$, if $v \in V_i$, and $\mu_i(v) = q$, if $v \notin V_i$. Thus, partitioning the rows of $D$ with a mixture model is exactly the same as recovering the clusters $U_i$ of $G$.

It is left to check that the conditions of Lemma 2 are satisfied. By assumption on the $V_j$, $\|\mu_i - \mu_j\|_2^2 \geq (p-q)^2\delta$ for $i \neq j$. Since we have $|U_i|$ samples from distribution $D_i$, the mixing weights are $w_i = |U_i|/m$ and $w_{\min} = \ell/m$. To apply Lemma 2, we must satisfy the inequality

$$(p-q)^2\delta > ck\sigma^2 \frac{m}{\ell} \left( \frac{m+n+m\lg m}{m} \right) = ck\sigma^2 \left( \frac{m+n+m\lg m}{\ell} \right).$$

By rearranging terms and noticing that $Cm\lg m \geq c(m + m\lg m)$ for large enough $C$, this is the inequality we required in the proposition (Equation 2). $\qquad\square$

# 4 Recovering the Right Clusters

This section presents an algorithm to recover the right clusters $V_j$ given the left clusters $U_i$. The algorithm is very simple: For each cluster $U_i$, $\tilde{V}_i$ consists of all vertices from $V$ which have "many" neighbors in $U_i$. We will show that the algorithm succeeds with high probability.

**High-Degree Thresholding Algorithm.** The input for the algorithm are $p, q$ and the clusters $U_1, \ldots, U_k$. For each cluster $U_i$, the algorithm constructs $\tilde{V}_i$ by adding all vertices $v \in V$ which have at least $\theta|U_i|$ neighbors in $U_i$, where we set

$$\theta = \lg\left(\frac{1-q}{1-p}\right)\left(\lg\left(\frac{p(1-q)}{q(1-p)}\right)\right)^{-1}. \tag{3}$$

**Proposition and Analysis.** We prove in Proposition 4 that for a fixed cluster $U_i$ of sufficiently large size, $V_i = \tilde{V}_i$ with probability $1 - O(n^{-2})$. A union bound implies that $\tilde{V}_i = V_i$ for all $i = 1, \ldots, k$ with high probability. In the proposition, we use the notation $D(p \parallel q)$ to denote the Kullback–Leibler divergence of Bernoulli random variables with parameters $p, q \in [0, 1]$, i.e., $D(p \parallel q) = p \lg(\frac{p}{q}) + (1-p) \lg(\frac{1-p}{1-q})$.

**Proposition 4.** *There exists a constant $C$ such that if $|U_i| \geq C \lg n / D(p \parallel q)$, then $\tilde{V}_i$ returned by the high-degree thresholding algorithm satisfies $V_i = \tilde{V}_i$ with probability at least $1 - O(1/n^2)$. The algorithm runs in time $O(|U_i|n)$.*

*Proof.* Consider a vertex $v \in V$. The vertex $v$ has an edge to $u \in U_i$ with probability $p$, if $v \in V_i$, and with probability $q$, if $v \notin V_i$. Let $Z_v$ be the random variable denoting the number of edges from $v$ to vertices in $U_i$; $Z_v$ is binomially distributed with $|U_i|$ trials and success probability $p$ (if $v \in V_i$) or $q$ (if $v \notin V_i$). To find out whether $v \in V_i$, we must decide whether $Z_v$ is distributed with parameter $p$ or $q$. If we decide for the correct parameter then the decision to include $v$ into $\tilde{V}_i$ is correct.

We make the decision for the parameter based on the likelihood of observing $Z_v$ edges incident upon $v$. Parameter $p$ is more likely if:

$$\frac{\binom{|U_i|}{Z_v}p^{Z_v}(1-p)^{|U_i|-Z_v}}{\binom{|U_i|}{Z_v}q^{Z_v}(1-q)^{|U_i|-Z_v}} = \left(\frac{p}{q}\right)^{Z_v}\left(\frac{1-p}{1-q}\right)^{|U_i|-Z_v} \geq 1.$$

Solving this inequality for $Z_v$ gives that one should decide for parameter $p$ if $Z_v \geq \theta|U_i|$, where $\theta$ is as in Equation 3.

The maximum likelihood approach above succeeds with probability at least $1 - O(1/n^3)$; this follows from [4, Theorem 6] if $|U_i| \geq C \lg n / D(p \parallel q)$, where $C$ is a sufficiently large constant. The probability for obtaining a correct result for *all* vertices $v \in V$ is at least $1 - O(1/n^2)$; this follows from a union bound. Conditioning on this event we obtain $V_i = \tilde{V}_i$. $\square$

# 5 Implementation

While so far we have been concerned with theory, we will now consider practice. The pseudocode of the algorithm we implemented is presented in Algorithm 1. As stated in Section 1, the algorithm performs two steps: (1) Recover the clusters $\tilde{U}_1, \ldots, \tilde{U}_k$ in $U$. (2) Recover the clusters $\tilde{V}_1, \ldots, \tilde{V}_k$ in $V$ based on the $\tilde{U}_i$. We call the algorithm pcv, which is short for *project, cluster, vote*.

While for step (2) we use exactly the algorithm discussed in Section 4, we made some changes for step (1). The main reason is that Mitra's algorithm discussed in Section 3 was developed in a way to give theoretical guarantees and not necessarily to give the best results in practice.

Instead, for step (1) we use an even simpler algorithm for recovering the clusters $\tilde{U}_i$: Project the biadjacency of $G$ on its first $k$ left singular vectors and then run $k$-means. This delivers better results and is conjectured to give the same theoretical guarantees as Mitra's algorithm (see [18] or [22]).

We implemented Algorithm 1 in Python. To compute the truncated SVD we used scikit-learn [23]. The source code is available in the supplementary material.

---

**Algorithm 1** The `pcv` algorithm

---

**Input:** $G$ a bipartite $m \times n$ graph, $k$, $p$, $q$

 1: **procedure** PCV($G$, $k$, $p$, $q$)
 2:      $D \leftarrow$ the $m \times n$ biadjacency matrix of $G$
 3:      $A \leftarrow$ rank $k$ SVD of $D$                                                    ▷ Step (1)
 4:      $\tilde{U}_1, \ldots, \tilde{U}_k \leftarrow$ the clusters obtained by running $k$-means on the rows of $A$
 5:      **for** $i = 1, \ldots, k$ **do**                                                   ▷ Step (2)
 6:          $\tilde{V}_i \leftarrow$ all vertices in $V$ with at least $\theta|U_i|$ neighbors in $U_i$, where $\theta$ is as in Equation 3

---

When developing the algorithm, we also tried using other clustering methods than $k$-means. However, none of them delivered consistently better results than $k$-means and the differences in the outputs were mostly minor. Hence, we do not study this further here.

We note that due to $k$-means, `pcv` is a randomized algorithm. On the synthetic graphs we will consider, this had almost no influence on the quality of the results. On real-world graphs, this randomness resulted in different clusterings in each run of the algorithm. However, some "prominent clusters" were always there and the computed clusters always had an interpretable structure.

**Parameters.** The parameters $p$ and $q$ are only used to compute the parameter $\theta$ from Section 4. We note that in practice it might be reasonable to pick a different threshold $\theta$ for each cluster depending on its sparsity; however, this was not done in this paper. In the supplementary material we present and evaluate a heuristic for how $p$ and $q$ (and, hence, $\theta$) can be estimated.

It suffices if $k$ is a sufficiently tight upper bound on its true value. `pcv` will not necessarily output exactly $k$ clusters; if $k$-means outputs less than $k$ clusters, then `pcv` will do the same. In practice it is sometimes handy to use different values for $k$ in the SVD and in $k$-means.

We further added a parameter $L \in \mathbb{N}$. In practice, often some of the $\tilde{U}_i$ returned by `pcv` are tiny (e.g., containing less than five vertices). To avoid creating too much output, we use the parameter $L$ to ignore all clusters $\tilde{U}_i$ of size less than $L$. In the experiments we always set $L = 10$.

## 6   Experiments

In this section, we practically evaluate the performance of `pcv`. Throughout the experiments our main objective will be to understand how well `pcv` can recover small clusters on the right side of the graph. In the synthetic experiments, we will be most interested in how small $p$ can be so that `pcv` can still recover clusters of size less than 10 on the right side of the graph. We picked real-world datasets from which we expect that they contain only very small clusters on the right side.

The experiments were done on a MacBook Air with a 1.6 GHz Intel Core i5 and 8 GB RAM. The source code and the synthetic data are provided in the supplementary materials.

**Algorithms.** `pcv` was compared with the `lim` algorithm by Lim, Wu and Xu [17], `message` by Ravanbakhsh, Póczos and Greiner [24], and the `lfm` algorithm by Rukat, Holmes and Yau [26]. For each of the algorithms, implementations provided by the authors were used. `message` and `lfm` are BMF algorithms (see Section 2).

When we report the running times of the algorithms, note that the quality of the implementations is incomparable. For example, `lim` is implemented in Matlab, `message` and `pcv` are purely implemented in Python and `lfm` is programmed in Python with certain subroutines precompiled using *Numba*.

### 6.1   Synthetic Data

Let us start by considering the performance of the algorithms on synthetically generated graphs. The graphs were generated as described in Section 1.

The ground-truth clusters $U_i$ and $V_i$ were picked in the following way. For each $U_i$, $\ell$ vertices were added to the (initially empty) left side of the graph. On the right side of the graph, we inserted $n$ vertices. Each of the $V_j$ consists of $r$ vertices which were picked uniformly at random from the

$n$ vertices. Due to the randomness in the graph generation, some of the $V_j$ will overlap and most of them will not. By *size of a cluster* we mean the number of vertices contained in the cluster.

When not mentioned otherwise, the parameters were set to $n = 1000$, $k = 8$, $\ell = 70$, and $m = \ell \cdot k$ (i.e., 1000 vertices on the right, 8 ground-truth clusters on both sides and left-side clusters of size 70). The size of the right-side clusters was set to $r = 8$. The parameters $p$ and $q$ were set depending on the dataset.

For each of the reported parameter settings, five random graphs were generated. The results that are reported in the following are averages over these datasets. When an algorithm was run multiple times on the same dataset, we report the best result on the right clusters of the graph.

During the experiments, all algorithms were given the correct parameters for $k$, $p$ and $q$ whenever the algorithms allowed this. For `lim` and `lfm` we optimized their parameters; we report this in the supplementary material.

**Quality Measure.** Consider the $k$ ground-truth clusters $U_1, \ldots, U_k$ and let $\tilde{U}_1, \ldots, \tilde{U}_s$ be the $s$ clusters returned by an algorithm. The *quality $Q$* of the solution $\tilde{U}_j$ is computed as follows. For each ground-truth cluster $U_i$, find the cluster $\tilde{U}_j$ which maximizes the Jaccard coefficient of $U_i$ and $\tilde{U}_j$. Then sum over the Jaccard coefficients for all ground-truth clusters $U_i$ and normalize by $k$. Formally,

$$Q = \frac{1}{k} \sum_{i=1}^{k} \max_{j=1,\ldots,s} J(U_i, \tilde{U}_j) \in [0, 1],$$

where $J(A, B) = |A \cap B| / |A \cup B|$ is the Jaccard coefficient. Higher values for $Q$ imply a better quality of the solution. E.g., if $Q = 1$ then the clusters $\tilde{U}_j$ match *exactly* the ground-truth clusters $U_i$.

We used the same quality measure for the clusters $V_i$. In the supplementary material we explain why decided against using the reconstruction error of the biadjacency matrix of $G$ as a quality measure.

**Varying $p$.** We start by studying how much the results of the algorithms are affected by destructive noise. To this end, we use varying values for $p = 0.2, 0.25, 0.3, 0.5, 0.75, 0.95$ and fix $q = 0.03$. The results are presented in Figures 1(a)–1(c).

We see that on both sides of the graph, `pcv` and `message` outperform `lfm` and `lim` for $p \leq 0.3$; for $p \geq 0.5$, `lim` picks up and delivers very good results.

In Figure 1(a) we see that on the left clusters, `pcv` and `message` deliver similar performances with `pcv` picking up the signal better for $p \geq 0.5$; the results of `lim` improve as $p$ increases and they are perfect for $p = 0.75, 0.95$; `lfm` always delivers relatively poor results.

For the right clusters the situation is similar with `message` having slight advantages over `pcv` for $p \leq 0.3$; `pcv` and `lim` deliver better results than `message` in settings with less noise ($p \geq 0.75$). It is interesting to observe that `pcv` already recovers the ground-truth clusters on the right side for $p \geq 0.4$ and even for $p = 0.3$ the results are of good quality.

The running times of the algorithms are reported in Figure 1(c). `pcv` is the fastest method with `lim` and `lfm` being somewhat slower. `message` is by far the slowest method and we see that when $p$ is small, `message` takes a long time until it converges.

**Varying sizes of the right clusters.** We now study how small the right clusters $V_i$ can get such that they can still be recovered by the algorithms. To this end, we vary the size of the right clusters and note that this corresponds to varying $\delta$ (for example, when all clusters are disjoint, $\delta$ is exactly twice the size of the right clusters).

Previously, we saw that `pcv`, `message` and `lim` did well at the recovery of right clusters of size 8 even for $p = 0.4$. We study this further by fixing $p = 0.4$, $q = 0.03$ and varying the size of the right clusters from 1 to 8. The results are reported in Figures 1(d)–1(f).

The results for clustering the left side of the graph are presented in Figure 1(d). We observe a clear ranking with `pcv` being the best algorithm before `message`; `lim` is the third-best algorithm and `lfm` is the worst.

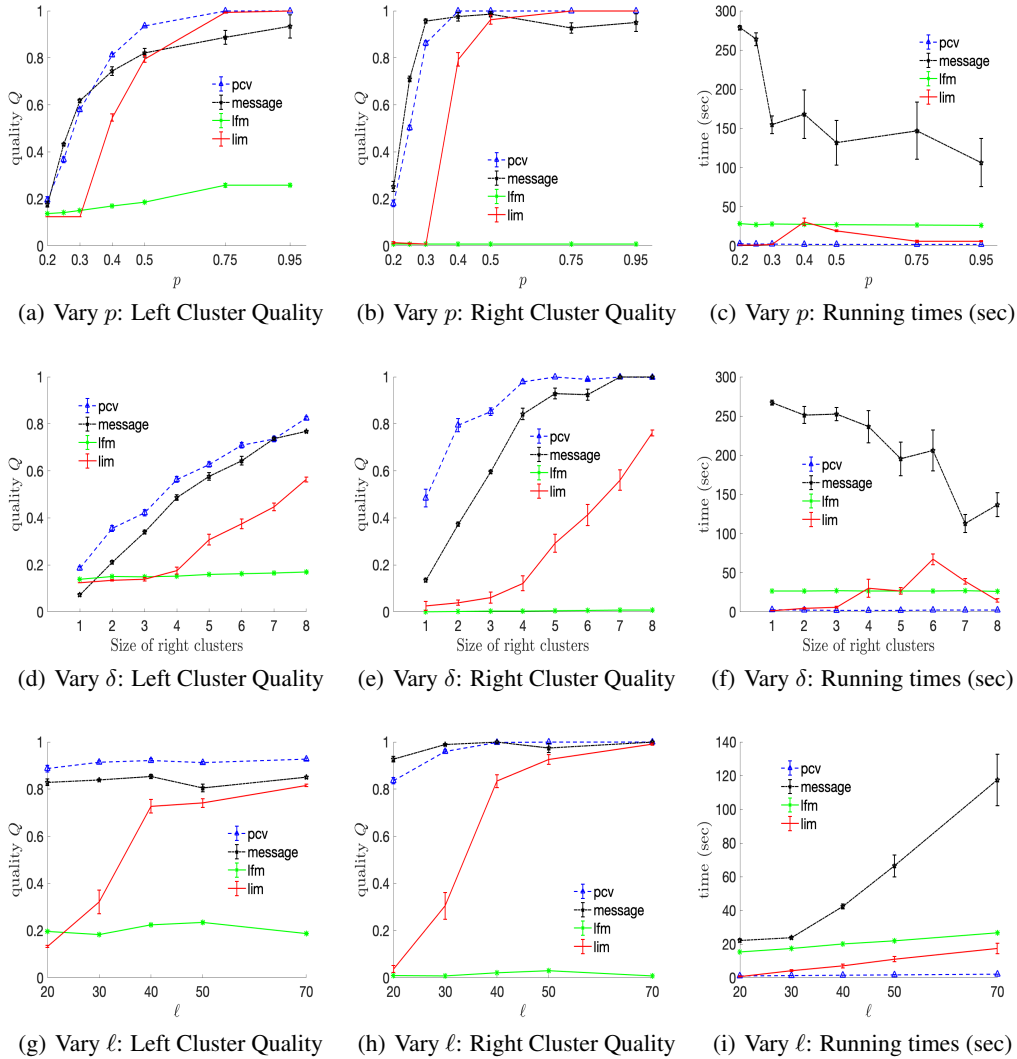

(a) Vary $p$: Left Cluster Quality    (b) Vary $p$: Right Cluster Quality    (c) Vary $p$: Running times (sec)

(d) Vary $\delta$: Left Cluster Quality    (e) Vary $\delta$: Right Cluster Quality    (f) Vary $\delta$: Running times (sec)

(g) Vary $\ell$: Left Cluster Quality    (h) Vary $\ell$: Right Cluster Quality    (i) Vary $\ell$: Running times (sec)

Figure 1: Results on synthetic data. Figures 1(a)–1(c) have varying $p$, Figures 1(d)–1(f) have varying sizes of the right clusters, Figures 1(g)–1(i) have varying $\ell$. Markers are mean values over five different datasets; error bars are one third of the standard deviation over the five datasets.

For the right side of the graph (Figure 1(e)) we observe that `pcv` outperforms `message` for ground-truth clusters sizes less than 7; even for clusters of sizes 2 and 3, `pcv` finds good solutions. The performance of `lim` improves as the cluster sizes grow.

The running times (Figure 1(f)) are similar to what we have seen before for varying $p$.

**Varying $\ell$.** We study how $\ell$, the size of the left clusters $U_i$, influences the results of the algorithms. We used values $\ell = 20, 30, 40, 50, 70$. The other parameters were fixed to $p = 0.5$, $q = 0.03$, $k = 8$ and the size of the right clusters was set to 8. The results are reported in Figures 1(g)–1(i).

On the left clusters, `pcv` is the best algorithm with `message` also delivering good results; the results of `lim` are of good quality for $\ell \geq 40$. On the right clusters, `message` is initially ($\ell \leq 30$) slightly better than `pcv` and for $\ell \geq 40$, `pcv` and `message` deliver essentially perfect results; `lim` finds good right clusters for $\ell \geq 40$. The running times are similar to what we have seen in previous experiments.

It is interesting and maybe even a bit surprising that even for $\ell = 20$, `pcv` and `message` can find very good clusters on the right side of the graph which only consist of 8 out of a 1000 vertices.

**Bad Parameters.** The supplementary material contains experiments in which we repeat the previous experiment with varying $\delta$ but where we run the algorithms with wrong parameters.

**Conclusion.** We conclude that `pcv` was very good at finding tiny clusters even with high destructive noise. In most cases, `pcv` delivered the solutions of highest quality and `pcv` was the fastest algorithm.

## 6.2 Real-World Data

`pcv` is qualitatively evaluated on two real-world datasets. Since the parameters required by `pcv` are not known, `pcv` was run with different parameters settings and the quality of the clusters was manually evaluated; the final setting of the parameters is reported for each dataset.

**Datasets.** The *BookCrossing* dataset[1] originates from Ziegler et al. [30]. It consists of users on the left side of the graph and books on the right side of the graph; if a user rated a book, there exists an edge between the corresponding vertices. The dataset was preprocessed so that all books read by less than 11 users and all users reading less than 11 books were removed. The resulting graph has 6195 users and 4958 books; the number of edges is 83550.

The *4News* dataset is a subset of the *20Newsgroups* dataset; it was preprocessed by Ata Kabán (see [19]). The data contains the occurrences of 800 words (right side of the graph) over 400 posts (left side of the graph) in four different Usenet newsgroups about cryptography, medicine, outer space, and christianity; for each newsgroup there are 100 posts. The graph has 11260 edges.

**Qualitative Evaluation.** *BookCrossing.* For the BookCrossing dataset, `pcv` was run with parameters $k = 20$, $\theta = 0.2$ and $L = 10$; `pcv` finished in less than 2 minutes.

`pcv` returns 12 user-clusters (i.e., on the left side of the graph) with size at least $L$. Out of these 12 user-clusters, 9 have a non-empty book-side (right side of the graph). The largest user-cluster contains 4268 vertices and has an empty book-side (right side). We will now discuss some of the clusters with non-empty right sides. All of those clusters have a natural interpretation.

The returned clusters mostly consist of books written by the same authors (as one would expect). Two clusters were consisting of the *Harry Potter* books by Joanne K. Rowling; the first cluster contained the five *Harry Potter* books that were published until 2004 (when the dataset was created) and contains 92 users, the other one consisted of the first three books of the series and contained 60 users. There is one cluster containing four books written by Anne Rice (64 users), one cluster containing seven books written by John Grisham (67 users), and one clusters containing 46 books written by Stephen King (12 users). `pcv` also returns two clusters containing a single book: *The Da Vinci Code* by Dan Brown (215 users) and *The Lovely Bones* by Alice Sebold (261 users).

*4News.* For this dataset we observe that it is useful to set the parameter $k$ in the SVD and in the call to $k$-means to different values. With this, we can obtain more general or more specific clusters: Setting the value $k$ for $k$-means to a smaller (larger) value, creates less (more) clusters on the left side of the graph. This will also make the right-side clusters more general (specific).

We used $k = 30$ for the SVD and $k = 50$ for $k$-means to obtain relatively specific clusters. The value of $k$ is so large, because the dataset contains many outliers that create a lot of left-side clusters of size 1. Further, we set $\theta = 0.3$, $L = 10$.

For each of the four newsgroups, `pcv` finds clusters. In total, `pcv` finds five clusters of which one has an empty right side (225 posts). The cluster (18 posts) returned for the cryptography newsgroup is *public, system, govern, encrypt, decrypt, ke(y), secur(ity), person, escrow, clipper, chip* (a *clipper chip* is an encryption device developed by the NSA). For the medicine newsgroups, `pcv` finds the cluster (24 posts) *question, stud(y), year, effect, result, ve, call, doctor, patient, medic, read, level, peopl(e), thing*. The cluster (19 posts) *concept, system, orbit, space, year, nasa, cost, project, high, launch, da(y), part, peopl(e)* explains the topics of the outer space newsgroup well. For the christian religion newsgroup we obtain the cluster (24 posts) *christian, bibl(e), read, rutger, god, peopl(e), thing*.

## Acknowledgements

I wish to thank the anonymous reviewers for for their helpful comments and for pointing out a heuristic to estimate parameters of the algorithm. I am grateful to my advisor Monika Henzinger for her support and for helpful discussions, to Pan Peng for valuable conversations about SBMs and to Pauli Miettinen and Jilles Vreeken for getting me interested in biclustering.

The author gratefully acknowledges the financial support from the Doctoral Programme "Vienna Graduate School on Computational Optimization" which is funded by the Austrian Science Fund (FWF, project no. W1260-N35).

## Footnotes

[1] http://www2.informatik.uni-freiburg.de/~cziegler/BX/

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
