[Supplementary Material]

# Supplementary Material For:
# Bipartite Stochastic Block Models with Tiny Clusters

Stefan Neumann

University of Vienna
Faculty of Computer Science
Vienna, Austria
stefan.neumann@univie.ac.at

## 1 Proof of Theorem 1

We prove Theorem 1. By Proposition 3, the clusters $U_i$ can be recovered with high probability. By Proposition 4 and the statement before the proposition, all $V_i$ can be recovered with high probability given the correct $U_i$. Now a union bound implies that both events happen simultaneously with high probability. Furthermore, the conditions of the propositions are satisfied because they are the same as those of Theorem 1.

## 2 Mitra's Algorithm

We present Mitra's algorithm [3]. The pseudocode of the algorithm is stated in Algorithm 1.

The main procedure of the algorithm is Cluster$(A, k)$ for which we describe some of its intuition. First, the algorithm splits $A$ into two parts $A_1$ and $A_2$ (this is a common trick in this line of work to ensure stochastic independence of the two parts and their projections). Then, for each part, the rank $k$ singular value decomposition (SVD) is computed (this step is supposed to reduce the noise) and the centers $\mu_i^*$ of each part are estimated using a standard clustering algorithm. To obtain the final clusters, the procedure Project$(A, \mu_1^*, \ldots, \mu_k^*)$ assigns each point $v$ to the closest center that was computed on the other part of the split matrix.

## 3 Approximate Left Clusters and Exact Right Clusters

In this section, we show that given a good *approximate* clustering of the left side of the graph, the clusters on the right side of the graph can still be recovered *exactly*.

Here, we are more interested in illustrating that results of the above type are possible; our goal is not to obtain the tightest possible analysis. Hence, to simplify the analysis, we make the following simplifying assumptions on the input data. We assume that each cluster $U_i$ contains exactly $\ell$ vertices. Thus, the left side of the graph contains $m = k\ell$ vertices. Furthermore, we assume that $q = Cp$ for a constant $C < 1$, i.e., $p$ and $q$ differ by a constant factor.

We will be working with the following definition of an approximate clustering. Let $U_1, \ldots, U_k$ be the ground-truth clusters. Then $\tilde{U}_1, \ldots, \tilde{U}_k$ is an $\varepsilon$-*approximate clustering* of $U_1, \ldots, U_k$ if there are at most $\varepsilon m$ misclassified vertices. More formally, we assume that

$$\sum_{i=1}^{k} |U_i \Delta \tilde{U}_i|/2 \le \varepsilon m = \varepsilon k\ell,$$

where $\Delta$ denotes the symmetric difference of two sets.

---
**Algorithm 1** Mitra's algorithm [3]
___
**Input:** $A \in \{0,1\}^{m \times n}$, $k$
**Output:** A clustering $\{P_1, \ldots, P_k\}$ for the samples
 1: **procedure** CLUSTER($A, k$)
 2:     Randomly split the rows of $A$ into equal-sized parts $A_1$ and $A_2$
 3:     $(\theta_1, \ldots, \theta_k) \leftarrow$ Centers($A_1, k$)
 4:     $(\nu_1, \ldots, \nu_k) \leftarrow$ Centers($A_2, k$)
 5:     $(P_1^1, \ldots, P_k^1) \leftarrow$ Project($A_1, \nu_1, \ldots, \nu_k$)
 6:     $(P_1^2, \ldots, P_k^2) \leftarrow$ Project($A_2, \theta_1, \ldots, \theta_k$)
 7:     **return** $\{P_1^1 \cup P_1^2, \ldots, P_k^1 \cup P_k^2\}$
 8: **procedure** CENTERS($A, k$)
 9:     $A^{(k)} \leftarrow$ rank-$k$-SVD of $A$
10:     Cluster the rows of $A^{(k)}$ into sets $P_1, \ldots, P_k$ using Euclidean distances (e.g., using $k$-means)
11:     $\mu_r^* \leftarrow \frac{1}{|P_r|} \sum_{v \in P_r} v$ for all $r = 1, \ldots, k$,
12:     **return** $(\mu_1^*, \ldots, \mu_k^*)$
13: **procedure** PROJECT($A, \mu_1^*, \ldots, \mu_k^*$)
14:     $P_r \leftarrow \emptyset$ for all $r = 1, \ldots, k$
15:     **for all** rows $v$ of $A$ **do**
16:         **for all** $r = 1, \ldots, k$ **do**
17:             **if** $|(v - \mu_r^*) \cdot (\mu_s^* - \mu_r^*)| < |(v - \mu_s^*) \cdot (\mu_r^* - \mu_s^*)|$ for all $s \neq r$ **then** Add $v$ to $P_r$
18:     **return** $(P_1, \ldots, P_k)$
---

We now state our main result which shows that given a good enough approximate clustering of the left side of the graph, the clusters $V_i$ can still be recovered exactly.

**Theorem 1.** *Suppose $\tilde{U}_1, \ldots, \tilde{U}_k$ is an $\varepsilon$-approximate clustering of $U_1, \ldots, U_k$ and let $q = Cp$ for a constant $C < 1$. Then there exist constants $D_1, D_2$ such that if (1) $\ell \geq D_1 \lg n/q$ and (2) $\varepsilon \leq D_2 p/k$, then there exists an algorithm which returns the clusters $V_1, \ldots, V_k$ with high probability.*

For the proof, we need the following well-known version of the Chernoff bound (see, e.g., Theorem 1.1 in Dubhashi and Panconesi [1]).

**Lemma 2.** *Let $X_1, \ldots, X_n$ be independent random variables in $[0, 1]$ and set $X = \sum_i X_i$. Then for $\varepsilon > 0$,*

$$\mathbf{Pr}\,(X > (1 + \varepsilon)\mathbf{E}\,[X]) \leq \exp(-\varepsilon^2 \mathbf{E}\,[X]\,/3),$$
$$\mathbf{Pr}\,(X < (1 - \varepsilon)\mathbf{E}\,[X]) \leq \exp(-\varepsilon^2 \mathbf{E}\,[X]\,/2).$$

*Proof of Theorem 1.* We are going to reuse the high-degree threshold algorithm from Section 4 of the paper. We will just use a different threshold which we specify later.

Fix any $i \in \{1, \ldots, k\}$. We show that given $\tilde{U}_i$, $V_i$ can be recovered with probability at least $1 - O(n^{-2})$. Using a union bound for the $k$ clusters, all $V_i$ are recovered with probability at least $1 - O(n^{-1})$.

To show that $V_i$ can be recovered with the desired probability, we show that with probability at least $1 - O(n^{-3})$ it can be decided whether $v \in V_i$ or $v \notin V_i$ for each $v \in V$. A union bound implies that $V_i$ is recovered exactly with probability at least $1 - O(n^{-2})$.

Observe that by the definition of an $\varepsilon$-approximate clustering, we have

$$\ell - \varepsilon k\ell \leq |\tilde{U}_i| \leq \ell + \varepsilon k\ell.$$

This implies that if $v \in V_i$, then $v$ has at least $\ell - \epsilon k\ell$ neighbors in $\tilde{U}_i$ to which it has an edge with probability $p$. Thus, the expected number of neighbors[1] of $v$ in $\tilde{U}_i$ is at least

$$\mu_1 = (\ell - \varepsilon k\ell)p.$$

On the other hand, if $v \notin V_i$, $v$ has at most $\ell$ neighbors in $V_i$ to which it has an edge with probability $q$ and at most $\varepsilon \ell k$ neighbors to which it has an edge with probability $p$. Thus, if $v \notin V_i$, its expected number of neighbors in $\tilde{U}_i$ is at most

$$\mu_2 = \ell q + \varepsilon \ell k p.$$

By the assumptions on $\ell$, $\varepsilon$, and $p$ and assuming that $D_2 \leq 1/2$, we obtain that

$$\mu_1 = \ell p (1 - \varepsilon k) \geq D_1 \lg n (1 - \varepsilon k) \geq D_1 \lg n / 2$$

and

$$\mu_2 = \ell q + \varepsilon \ell k p \geq \ell q \geq D_1 \lg n.$$

Let $\alpha < 1$, $\beta > 1$ be constants such that $\alpha/\beta > C$. Now setting $D_1$ to a large enough constant and applying Lemma 2, we obtain that the following two events occur with probability at least $1 - \exp(-3 \lg n) = 1 - O(n^{-3})$: (1) If $v \in V_i$, then $v$ has at least $\alpha \mu_1$ neighbors in $\tilde{U}_i$. (2) If $v \notin V_i$, then $v$ has at most $\beta \mu_2$ neighbors in $\tilde{U}_i$.

Now observe that with our choice of parameters we have that $\alpha \mu_1 > \beta \mu_2$ since:

$$\alpha \mu_1 > \beta \mu_2$$
$$\iff \quad \alpha \ell p (1 - \varepsilon k) > \beta \ell (q + \varepsilon k p)$$
$$\iff \quad \alpha p (1 - \varepsilon k) > \beta (q + \varepsilon k p)$$
$$\iff \quad \alpha p - (\alpha + \beta) \varepsilon k > \beta q$$
$$\iff \quad \alpha p - \beta q > (\alpha + \beta) \varepsilon k$$
$$\iff \quad \varepsilon < \frac{\alpha p - \beta q}{(\alpha + \beta) k} = \frac{p(\alpha - \beta C)}{(\alpha + \beta) k}$$

Observe that $\alpha - \beta C$ is always positive since we assumed that $\alpha/\beta > C$. Furthermore, the above inequality can be satisfied by making $D_2$ sufficiently small.

We conclude that the following algorithm succeeds with probability at least $1 - O(n^{-3})$ for each $v$. Include $v$ in $V_i$ if $v$ has at least $\alpha \mu_1$ neighbors in $\tilde{U}_1$. Otherwise, do not insert $v$ into $V_i$. $\qquad\square$

## 4 Heuristic for Estimating $p$ and $q$

In this section, we discuss a heuristic for estimating the parameters $p$ and $q$ of `pcv`. This heuristic was generously pointed out by one of the anonymous reviewers and we are grateful for this contribution.

Note that the first step (computing SVD and applying $k$-means) of `pcv` does not require knowledge about the parameters $p$ and $q$. Thus, the recovery of the left clusters does not need to be changed. We only need to argue how the second step of the algorithm needs to be adjusted when the parameters $p$ and $q$ are not known.

Consider the second step of the algorithm, i.e., the high-degree thresholding algorithm. Recall that for a given transaction cluster $U_i$, the algorithm sets $V_i$ to the set of vertices containing all $v \in V$ with at least $\theta |U_i|$ neighbors in $U_i$. Here, computing $\theta = \theta(p, q)$ requires knowledge about $p$ and $q$.

Now suppose we obtain a set $V_i$ which was computed in the previously specified way for parameter $\theta = \theta(p, q)$.

By assumption of the SBM model, we know that in a case without noise (i.e., $p = 1$, $q = 0$), all vertices in $V_i$ would have $|U_i|$ edges to vertices in $U_i$ and 0 edges to vertices outside $U_i$ (here, we assume that the $V_i$ do not overlap). In the noiseless setting, there would be a total of $|V_i| \cdot |U_i|$ edges from vertices in $V_i$ to vertices in $U_i$.

Now, we can also compute the number of edges which are present in the random graph. To this end, let $|E(U_i, V_i)|$ denote the number of edges between vertices in $U_i$ and $V_i$ and let $|E(U \setminus U_i, V_i)|$ denote the number of edges between vertices in $U \setminus U_i$ and $V_i$.

Now we can *estimate* the parameters $\hat{p}$ and $\hat{q}$ from the cluster $V_i$ that was computed from $\theta(p, q)$. For this purpose, we set

$$\hat{p} = \frac{|E(U_i, V_i)|}{|V_i| \cdot |U_i|} \quad \text{and} \quad \hat{q} = \frac{|E(U \setminus U_i, V_i)|}{|V_i| \cdot |U \setminus U_i|}.$$

Note that here $\hat{p}$ ($\hat{q}$) is simply the fraction of edges which should (not) have been there in the noiseless setting and which were observed in the random graph.

Obviously, if $p$ and $q$ were the correct parameters for generating $V_i$, then we should have $p \approx \hat{p}$ and $q \approx \hat{q}$. In particular, $|p - \hat{p}| + |q - \hat{q}|$ should be small.

This gives raise to the following heuristic algorithm for estimating good parameters $p$ and $q$: Let $\mathcal{P}$ be a set of candidates for $p$ and let $\mathcal{Q}$ be a set of candidates for $q$. Now iterate over all tuples $(p, q) \in \mathcal{P} \times \mathcal{Q}$ such that $p > q$. For each such tuple, generate the set $V_i$ with parameter $\theta(p, q)$ using the high-degree thresholding algorithm. Given the set $V_i$, estimate $\hat{p}$ and $\hat{q}$ as described above. Of all the tuples, pick the one which minimizes the objective function $|p - \hat{p}| + |q - \hat{q}|$.

We wish to point out that the above procedure can be optimized using grid search instead of iterating over all tuples in $\mathcal{P} \times \mathcal{Q}$.

We experimentally evaluate the heuristic in Section 5.4.

# 5 Experiments

## 5.1 Algorithms

We provide some details of how we executed `lim` and `lfm` on the synthetic datasets.

`lim` takes as input a weight matrix $W$ which is a weighted version of the biadjacency matrix $B$ of the graph. After a correspondence with the authors of [2], we set

$$W = \lg\left(\frac{p}{q}\right) B + \lg\left(\frac{1-p}{1-q}\right)(1 - B).$$

As output, `lim` returns a denoised version of the data. To obtain the left and right clusters of the graph, we applied $k$-means first to the rows of the output and then 2-means to the columns of the submatrices of the output; this is similar to what was reported in [2]. Further, `lim` has a parameter $\lambda$ which [2] set to $\sqrt{2n}$ (this is $44.7$ in our setting); we have run the algorithm with parameter $\lambda = 20, 25, 30, 35, 40, 45$ as sometimes this gave better results.

For the `lfm` algorithm we inverted the data (i.e., we ran the algorithm on the complement graph) to improve its performance and we fixed the value for $\lambda$ to $0.5$ for the first 100 iterations. Furthermore, we set the number of latent dimensions to $k$. This procedure was suggested in a correspondence with one of the authors of [4].

Since the results of the `lfm` algorithm depended heavily on the randomness of the algorithm, we ran the algorithm 10 times on each dataset; all other algorithms were run once.

When the algorithms returned fractional values (i.e., in the interval $(0, 1)$), we rounded them to 0/1 with threshold $0.5$.

## 5.2 Quality Measure

We wish to further motivate why we used the quality measure $Q$ instead of using the reconstruction error.

Let us first the define the reconstruction error. Let $B$ be the biadjacency matrix of the input graph $G$. Then for the outputs $\tilde{U}_i$, $\tilde{V}_i$ of an algorithm, define a matrix $A$ by setting $A_{uv} = 1$ iff there exists an $i$ such that $u \in \tilde{U}_i$ and $v \in \tilde{V}_i$. Now the reconstruction error is given by $||A - B||_2^2$.

The main advantage of the reconstruction error is that it does not require knowledge about the ground-truth clustering. Thus, it can be easily computed also on real-world datasets.

However, it has two major drawbacks for our purposes.

First, the reconstruction error does not allow us to understand how well the the algorithms perform on each side of the graph. With the quality measure $Q$, this is possible.

Second, in our experiments we considered scenarios with very high destructive noise. E.g., for parameter $p = 0.4$ it is more likely that a edge $(u, v)$ from $u \in U_i$ to $v \in V_i$ is *not* present than

| (a) Vary $\delta$: Left Cluster Quality | (b) Vary $\delta$: Right Cluster Quality | (c) Vary $\delta$: Running times (sec) |

Figure 1: Results on synthetic data. Figures 1(a)–1(c) have varying sizes of the right clusters and the algorithm were executed with wrong parameters. Markers are mean values over five different datasets; error bars are one third of the standard deviation over the five datasets.

that is this present in the graph. This implies that *the reconstruction error of the empty graph has lower reconstruction error than the ground-truth clustering*. Thus, in settings with very small $p$, the reconstruction error favors solutions which are very dissimilar from the ground-truth data. The quality measure $Q$ does not have this drawback.

## 5.3 More Synthetic Data

We provide more experiments on synthetically generated data.

**Varying right cluster sizes with wrong parameters.** We repeated the experiment for varying sizes of the right clusters that we reported in the main text but this time we gave the algorithms incorrect values for their parameters. This should affect `pcv`, `message` and `lim` since they take the most parameters. The results can be seen in Figures 1(a)–1(c).

Recall: `pcv` and `lim` take as parameters $p, q, k$. `message` takes as parameters, $p, q, k$ and the sizes of the clusters on the left sides and the right sides; we were generous and provided `message` with the correct values for the cluster sizes.

The true parameter values were $p = 0.4$, $q = 0.03$ and $k = 8$. We gave the algorithms the incorrect values $\tilde{p} = 0.6$, $\tilde{q} = 0.01$ and $\tilde{k} = 12$.

For the left clusters we see that for clusters of size 1 and 2, `pcv` is better than `message`, for sizes 3 and 4 they are on par and after that `message` is better. For the right side clusters we observe that until size 4 `pcv` is better than `message` and after that they are on par with perfect or almost perfect results. For `lim` we see that its performance improves as the cluster sizes grow.

Compared with the results with the correct parameters from the main text, the performance of all algorithms was relatively robust. `pcv`'s performance on the left clusters decayed while on the right clusters its results were relatively stable. For `message` the results were very robust and its performance on the right clusters even slightly improved; the latter might be down to much higher running times as reported in Figure 1(c) (apparently the algorithm takes a longer time to converge when run with incorrect parameters). For `lim` we observe that the change in the parameters has only a small influence on its results.

The fact that the performance of `pcv` decayed on the left clusters is down to the following two facts: (1) Rank 12 SVD picks up more noise than the rank 8 SVD. (2) When clustering the left side into 12 clusters instead of 8, $k$-means will simply partition the left side into too many sets. Due to (2), each left cluster is smaller than with the correct value for $k$. This causes the inference of the right side clusters of size at least 5 to be slightly less robust than when the algorithm is run with the correct parameters. However, note that for the right clusters these effects are only minor.

| (a) Vary $p$: Right Cluster Quality | (b) Vary $\delta$: Right Cluster Quality | (c) Vary $\ell$: Right Cluster Quality |

Figure 2: Results on synthetic data for `pcv` and `pcv` with the heuristic from Section 4. All plots report the quality for the recovery of the right clusters. Figure 2(a) has varying values for $p$, Figure 2(b) has varying sizes of the right clusters and Figure 2(c) has varying values for $\ell$. Markers are mean values over five different datasets; error bars are one third of the standard deviation over the five datasets.

## 5.4 Evaluation of the Heuristic

We evaluate the heuristic which was presented in Section 4 and compare it with the version of `pcv` which knows all parameters. For the heuristic, we set the candidates for $p$ to $\mathcal{P} = \{0.3, 0.35, 0.4, \ldots, 0.95\}$ and the set of candidates for $q$ to $\mathcal{Q} = \{0.01, 0.02, 0.03, \ldots, 0.1\}$.

We used the same synthetic datasets that we already discussed in the main text of the paper. We only report the results on the right clusters since the results on the left are exactly the same (because the the clustering of the left side of the graph did not change). The results are stated in Figure 2.

We see that (surprisingly) for varying values of $p$ (Figure 2(a)), the heuristic version of `pcv` is slightly better than `pcv` with the correct parameters for very small values of $p$. For varying sizes of the right clusters (Figure 2(b)), the heuristic is slightly worse than the version of `pcv` which knows the correct parameters. For varying $\ell$ (size of the left clusters) in Figure 2(c), the heuristic is visibly worse. The latter is perhaps to be expected because for smaller left clusters the inference on the right clusters might overfit easily when the heuristic is used.

Note that in all experiments the standard deviation of the qualities returned by the heuristic is much higher. This is not surprising since the solution returned by the heuristic heavily depends on the randomness in the data (whereas `pcv` with the correct parameters mainly requires a good clustering of the left side of the graph).

We conclude that when the left clusters and the right clusters are both large enough, then the heuristic delivers results of good quality.

## Footnotes

[1] Note that if $v \in V_i$, $\tilde{U}_i$ might contain vertices to which $v$ has edges with probability $q$. However, we can safely ignore these when computing the the lower bound on the expected number of neighbors of $v$ in $\tilde{U}_i$.