[Reviews · NeurIPS 2018]

Reviewer 1



This paper introduces and analyzes a simple algorithm for bipartite block model clustering in which clusters can be very small. The model in which the work is situated is a natural variation of a blockmodel for bipartite random graphs which allows the clusters on the "right" to overlap. The model involves a bijection between the clusters on the left and the clusters on the right. Two probabilities must be specified: the probability that an edge exists between two nodes whose blocks are in bijection, and the probability that an edge is generated between two nodes whose blocks are not in bijection (though it is likely that this constraint can be relaxed). The algorithm is simple: first, cluster the nodes on the left by applying a mixture model approach to the rows of the biadjacency matrix. Next, having determined the clusters on the left, assign to each the nodes on the right which have sufficiently many edges to it. To analyze the first part of the algorithm, the authors make use of the results of Mitra concerning mixture models. The authors prove that -- quite interestingly -- the algorithm is able to recover right clusters which are O(n^eps) small, which represents a significant improvement over previous analyses. This is perhaps surprising due to previous analyses which proved a sqrt(n) lower bound on cluster size, however (as the authors note) those analyses assumed that the bipartite clusters were square. In contrast, the current paper allows the left cluster and right cluster to be of vastly different sizes. This relaxation is quite natural and provides for a smooth tradeoff -- very small right clusters can be recovered, provided that the left cluster is correspondingly large. Ultimately, I found the paper clear and interesting. The fact that such small clusters can be recovered is intriguing, though it would be interesting to investigate whether the smooth tradeoff mentioned is indeed optimal.

Reviewer 2



This paper is of big significance in studying the clustering with tiny clusters. It states the conditions under which the proposed algorithm succeeds, and provides sufficient theoretical proofs to these statements. Strengths: -This paper is well organized and presented. The expression are clear making the reader easily follow this paper. -The originality of this paper is good, which gives the definition of “tiny cluster” and points out the importance of clustering these tiny clusters. -The techniques and derivations in the paper is reasonable and convince me. Weaknesses: -There is still several typos, such as “of” in line 127 is needless, “iff” in line 136 is a spelling mistake. -Some notations in the paper are unclear, such asΟ(.), Ω(.)and Θ(.). -In your experiment, how to understand the definition of “planted cluster”? Is it a typo in the expression “70 left-side clusters of size 70”? Since you have said 8 planted cluster,it is confused that you mean the number of clusters is 70 or each size of data in any cluster is 70. -How to understand the significance of experimental results in Figure 1 (d)(e)(f)? In the paper, the parameter delta is defined as “the smallest difference between any two right-side clusters” instead of “the size of right cluster”. So, why you study the ability of the proposed method in recovering tiny clusters under varying delta, rather than l (which is defined as the smallest cluster on left side)?

Reviewer 3



This paper considers a bipartite Stochastic Block Model (SBM) that allows for overlapping right clusters and proposes a simple two step algorithm to estimate the left and right clusters with provable guarantees. The main selling point is that this algorithm, unlike previous ones, can detect small clusters (of size (n^eps), for any eps > 0, as opposed to Omega(n^0.5) in previous works). This is practically useful because in many real bipartite networks the right side can have tiny clusters. The authors analyze two real-world examples to illustrate this. Comparisons with other algorithms on synthetic networks are also given. I found the paper interesting and believe that it is a good contribution to the literature. Some comments/suggestions for the authors: 1. Theorem 1 concerns strong recovery, and so needs the graph to have unbounded average degree. A discussion about the sparse case would be welcome. Does the pcv algorithm (or Mitra's algorithm plus your second step) return clusters "correlated with" the ground truth in that case? 2. For estimating theta a naive approach would be to start with a pilot estimate (e.g., grid search) to get the clusters first and then estimate p and q by within (i.e. between U_i and V_i for all i) and between (i.e. between U_i and V_j for all i \ne j) block averages of edges, and finally re-estimate theta using these estimates of p and q, and run the second step of your algorithm. This may be repeated until "convergence" (when the estimate of theta essentially stops changing, which may or may not happen). 3. Page 7, line 248: "used the quality" should be "used the same quality"? 4. Page 8, line 301: "Like this" should be "With this". 5. Page 8, line 314: "(24 posts) is christian" should be "(24 posts) christian".